# Sifting composite from elementary models at FCCee and CePC

G. Cacciapaglia[1,2,*], A. Deandrea[1,2,†], A.M. Iyer[3,‡] and A. Pinto[1,2,4§]

[1] *University of Lyon, Université Claude Bernard Lyon 1, F-69001 Lyon, France*
[2] *Institut de Physique des 2 Infinis de Lyon (IP2I),*
*UMR5822, CNRS/IN2P3, F-69622 Villeurbanne Cedex, France*
[3] *Department of Physics, Indian Institute of Technology Delhi, New Delhi-110016, India*
[4] *IRFU, CEA, Université Paris-Saclay, 91191 Gif-sur-Yvette, France*

New Physics models with either an elementary or composite origin are often associated with a similar imprint in a direct search at colliders, case in point being the production of a light pseudoscalar in association with a monochromatic photon from the decay of a Z boson at future $e + e-$ colliders. We exploit the correlation between the discovery of a signal in the Z decays and electroweak precision measurements as a tool to distinguish a composite model from an elementary scalar one. Our results offer an appealing and rich physics case for future colliders and demonstrate how a lepton collider at the Z mass can be a discovery machine for new physics in the Higgs sector.

## I. INTRODUCTION

The mechanism underlying the electroweak (EW) symmetry breaking in the Standard Model (SM) has been the subject of intense inquisition. The SM Higgs sector, in fact, provides a neat description via an elementary scalar field with an ad-hoc potential [1], while also suffering from the well-known radiative instability of the EW scale (naturalness problem). One of the most attractive possibilities follows from the analogy with chiral symmetry breaking in Quantum Chromodynamics (QCD). In such scenarios, the gauge group of the SM is embedded in a global symmetry group $G$, spontaneously broken via a fermion condensate [2] to a subgroup $H$, which only contains the generator of electrodynamics (QED). The underlying dynamics requires a new confining group with fermions charged under it (and also carrying EW charges). Henceforth, several Goldstone bosons emerge, sitting in the coset space $G/H$. Two extreme possibilities can be realised for the breaking of the EW symmetry:

A) In Technicolor-like [2–4] scenarios the breaking of $G$ also results in the EW breaking, hence the Higgs vacuum expectation value (vev) $v$ is replaced by the Goldstone decay constant $f = v$;

B) In composite Higgs scenarios [5–8], the Higgs doublet emerges as a pseudo-Nambu-Goldstone boson (PNGB) and the EW symmetry is broken via vacuum misalignment [5], leading to $f \gg v$.

Note that a misalignment model can always be rotated to its Technicolor limit, while the viceversa is not always possible [9]. One generic consequence of this mechanism is the emergence of additional pseudo-scalars, which are potentially lighter that the Higgs boson. In many cases, their leading-order linear couplings to SM particles stem from the Wess-Zumino-Witten (WZW) topological anomaly [10, 11], which yields couplings to the EW gauge bosons in the form

$$\mathcal{L}_{\text{WZW}} \supset a \left( g^2 \frac{C_W}{\Lambda} W_{\mu\nu} \tilde{W}^{\mu\nu} + g'^2 \frac{C_B}{\Lambda} B_{\mu\nu} \tilde{B}^{\mu\nu} \right) , \quad (1)$$

where $a$ is a light composite pseudoscalar singlet under the EW symmetry, $\Lambda$ is the cut-off of the effective theory, while $W, B$ are the $SU(2)_L$ and $U(1)_Y$ gauge bosons, respectively. The coefficients $C_{W,B}$ are determined by the underlying completion of the model. The pseudoscalar $a$ could either be embedded in models of composite Higgs [7, 12–14] or be a generic axion-like particle (ALP) [15, 16]. In either case, we are interested in scenarios where $a$ lacks leading-order couplings to gluons and SM fermions. Hence, the couplings to fermions are loop-induced [17] and are entirely determined by the WZW couplings in Eq. (1), making the model highly predictive. Thus, the only free parameters are its mass $m_a$ and the decay constant $f_a \sim f$, the latter appearing in the expression for $C_W/\Lambda$ and $C_B/\Lambda$.

The run of a future $e + e-$ collider at the $Z$ mass energy offers a bright prospect for the discovery of these states if $m_a < m_Z$. This includes the prospected Tera-Z run of the FCC-ee project [18, 19] as well as the planned Giga-Z run at the CePC project [20, 21]. The analysis in Ref.[22] considered the $Z$ portal production of the pseudoscalar $a$ in association with a monochromatic photon, $Z \to a \gamma$. Depending on the mass and the decay constant $f$, the pseudoscalar decays promptly, with a displaced vertex (long lived case) or outside the detector, manifesting itself as missing energy. The entire analysis was duly classified on the basis of the presence of a coupling to photons: In the photophobic case ($C_B = -C_W$), the decays are due to the loop-induced couplings to fermions, preferably to bottom quarks, leading to larger parameter space with long lived or missing energy signatures; In the remaining photophilic cases, the decay is dominantly into a pair of photons by means of the WZW vertex in Eq.(1).

________

[*] g.cacciapaglia@ip2i.in2p3.fr
[†] deandrea@ipnl.in2p3.fr
[‡] iyerabhishek@physics.iitd.ac.in
[§] andres.eloy.pinto.pinoargote@cern.ch

arXiv:2211.00961v1 [hep-ph] 2 Nov 2022

Of particular interest are cases with long lived signatures, as the presence of a displaced vertex in association with a monochromatic photon is characterised by negligible SM backgrounds. Given that the long lifetimes of these composite pseudoscalars were due to the absence of leading order couplings to gluons and fermions, one was led to wonder if such a signal is an incontrovertible smoking gun of compositeness.

In this work we give a closer look into this hypothesis and explore the possibility of obtaining the same signature from elementary scalar extensions of the SM. We construct a mockup model with elementary (pseudo)scalars and vector-like leptons (VLLs). To distinguish the elementary model from the composite case, provided the same signature and rates, we consider corrections to EW precision measurements as a discriminator. In fact, a future $e + e-$ collider will provide huge improvements in the tests of the properties of the EW gauge bosons, provided a substantial improvement in the theoretical precision is also achieved. Our results offer a strong case for the use of $e + e-$ machines, like FCCee and CePC, as discovery tools via a strong synergy of direct searches and EW precision measurements. This example also provides a powerful probe of the origin of the EW symmetry breaking in the SM.

The paper is organised as follows: In Section II we briefly summarise the properties of the light composite pseudoscalars. In Section III we detail the salient features of the mockup model, consisting of a complex singlet scalar extension to the SM plus heavy VLLs. In Section IV we show how the discovery of the $Z \to a\,\gamma$ process plus EW precision tests provides a strong discrimination power. Finally, we draw our conclusions in Section V.

## II. COMPOSITE ALPS

Composite Higgs models with an underlying gauge-fermion description always feature additional PNGBs, besides the Higgs and the Goldstones eaten by the $W$ and $Z$ bosons [7, 23, 24]. For example, the minimal model features the symmetry breaking pattern $SU(4) \to Sp(4)$ and has an additional pseudoscalar singlet [9, 12]. In general, we are interested in light pseudoscalars that only couple to the EW gauge bosons via the WZW terms in Eq. (1). The coefficients depend on the properties of the underlying gauge theory, and read (at leading order in $v/f$):

$$\frac{C_{W/B}}{\Lambda} = \frac{d_\psi}{64\sqrt{2}\pi^2} \frac{c_{W/B}}{f}\,, \qquad (2)$$

where $d_\psi$ is the dimension of the underlying fermion representation, and $c_{W/B}$ are group theory factors related to the coset of the specific model. A photophobic $a$ is, therefore, present if $c_B = -c_W$, situation that occurs in cosets of the form $SU(N_f)/Sp(N_f)$ and $SU(N_f)^2/SU(N_f)$. In the minimal cases, $N_f = 4$, we have $c_W = -c_B = 1$.

For $a$ lighter than the $Z$ boson, only couplings to photons are relevant for the phenomenology, which, from Eq. (1), read

$$\mathcal{L}_{\text{WZW}} \supset a\, e^2 \left( \frac{C_{\gamma\gamma}}{\Lambda} A_{\mu\nu}\tilde{A}^{\mu\nu} + \frac{2}{s_W c_W}\frac{C_{\gamma Z}}{\Lambda} A_{\mu\nu}\tilde{Z}^{\mu\nu} + \dots \right)\,, \qquad (3)$$

where $C_{\gamma\gamma} = C_W + C_B$ and $C_{\gamma Z} = c_W^2 C_W - s_W^2 C_B$. Here, $s_W$ and $c_W$ are the sine and cosine of the Weinberg angle, respectively. In the photophobic case, $C_{\gamma\gamma} = 0$ and $C_{\gamma Z} = C_W$. The BR($Z \to a\gamma$) can, therefore, be directly related to the WZW coefficient (and the compositeness scale $f$) by

$$\text{BR} = \frac{8\pi\alpha\, m_Z^5}{3 s_W^2 c_W^2 \Gamma_Z} \left(1 - \frac{m_a^2}{m_Z^2}\right)^3 \left(\frac{C_W}{\Lambda}\right)^2\,. \qquad (4)$$

Note that, for fixed $m_a$ and using Eq. (2), $f$ can be related to the BR by inverting the above equation.

## III. ELEMENTARY MOCK-UP: A COMPLEX SCALAR MODEL

Complex singlet scalar extensions of the SM (cxSM) have been the subject of extensive investigations, ranging from their implications for collider phenomenology to cosmological considerations [25–33]. In this work we consider one complex singlet $\mathbb{S} = (S + i\,a)/\sqrt{2}$ where, with abuse of notation, we call $a$ the pseudoscalar component. We assign to $\mathbb{S}$ the most general and simplest renormalisable scalar potential with approximate global $U(1)$ and discrete $Z_2$ symmetries. Hence, the scalar potential of the cxSM we consider, including the Higgs doublet field $H$, reads [29–33]:

$$V(H, \mathbb{S}) = \frac{m^2}{2} H^\dagger H + \frac{\lambda}{4}(H^\dagger H)^2 + \frac{\delta_2}{2} H^\dagger H\, |\mathbb{S}|^2$$
$$+ \frac{b_2}{2}|\mathbb{S}|^2 + \frac{d_2}{4}|\mathbb{S}|^4 + \frac{|b_1|e^{i\phi_{b_1}}}{4}\mathbb{S}^2 + |a_1|e^{i\phi_{a_1}}\mathbb{S}, \quad (5)$$

where $m$ and $\lambda$ are the usual SM Higgs potential parameters. The last two complex couplings, $b_1$ and $a_1$ break explicitly the approximate symmetries $U(1)$ and $\mathbb{Z}_2$, respectively, and can be considered small compared to the other parameters in the potential. Following Eq. (5), both $H$ and $\mathbb{S}$ develop a vev, where $\langle S \rangle = v_S$. The scalar spectrum consists of two scalars, $h_{1,2}$, stemming from the mixing between the SM Higgs and $S$, and a pseudoscalar $a$. Either $h_1$ or $h_2$ corresponds to the 125 GeV Higgs boson discovered by ATLAS and CMS, and a crucial parameter here is the mixing angle $\phi$ which is constrained by the measurement of the Higgs couplings and also controls the EW precision tests in this model, as we will see. The pseudoscalar acquires a mass

$$m_a^2 = |b_1| + \frac{\sqrt{2}|a_1|}{v_S}\,, \qquad (6)$$

hence proportional to the couplings that break the global symmetries. As such, $a$ is a PNGB and it is enabled to be arbitrarily light. As we are interested in the production of these particles through the decay of a Z boson, we can ensure all possible values of the mass between 1 and 90 GeV. Note that this low mass is not in contradiction with having $v_S \gg v$, so that $m_{h_2} \gg m_{h_1} = 125$ GeV.

We consider next the the couplings of the new scalars to SM particles. The scalar $h_2$ inherits them from the mixing to the SM Higgs boson, while the pseudoscalar $a$ has no tree level couplings, being a gauge singlet. To generate couplings of a single $a$ to the EW gauge bosons, similar to the composite WZW case, we introduce VLLs $\Psi_j$ in different representations of the EW gauge symmetry, and introduce the following Lagrangian:

$$\mathcal{L}_\Psi = \sum_j \bar{\Psi}_j i \not{D} \Psi_j - \mu_j \bar{\Psi}_j \Psi_j - y_j \, \mathbb{S} \, \bar{\Psi}_j \Psi_j . \quad (7)$$

Note that the presence of both a mass term $\mu_j$ and a coupling $y_j$ breaks the global symmetries in the scalar sector. At one loop, the VLLs $\Psi_j$ generate couplings to the EW gauge bosons in the form of Eq. (1), with coefficients

$$\frac{C_W}{\Lambda} = \frac{1}{8\pi^2} \sum_j \, T(R_j) \, \frac{y_j}{\sqrt{2}m_j} \, I_0\left(\frac{m_a}{m_j}\right) , \quad (8)$$

$$\frac{C_B}{\Lambda} = \frac{1}{8\pi^2} \sum_j \, Y_j^2 \, \frac{y_j}{\sqrt{2}m_j} \, I_0\left(\frac{m_a}{m_j}\right) , \quad (9)$$

where $m_j = \mu_j + y_j v_S/\sqrt{2}$ is the mass of the multiplet, $R_j$ and $Y_j$ its SU(2) representation and hypercharge, respectively, and $I_0$ a loop function that tends to a constant for large VLL mass, $I_0(0) = 1/2$.

To impose photophobic couplings, so that $C_\gamma = C_W + C_B = 0$, requires non-trivial constraints on the parameters of the VLL sector. First of all, if the mass is only due to the $\mathbb{S}$ vev, i.e. $m_j = y_j v_S/\sqrt{2}$, then this condition cannot be achieved without forbidding all other couplings to the EW gauge bosons. This is more illuminating by writing explicitly $C_\gamma$:

$$\frac{C_\gamma}{\Lambda} = \frac{1}{8\pi^2} \sum_j \, \frac{y_j}{\sqrt{2}m_j} \, \mathrm{Tr}\left[Q_\Psi^2\right] \, I_0\left(\frac{m_a}{m_j}\right) = 0. \quad (10)$$

For $\mu_j = 0$, then $\frac{y_j}{m_j} = \sqrt{2}/v_S$ and the coefficient is always positive, unless all VLLs are gauge singlets. The only way to ensure a photophobic elementary $a$ is to require a cancellation due to the signs of the Yukawas $y_j$ for $\mu_j \neq 0$. For concreteness, in the following we consider two scenarios where the cancellation can occur.

**Case A.** We introduce a doublet $\Psi_1$ and a singlet $\Psi_2$. A minimal choice for the hypercharges that avoids semi-integer electric charges is $Y_1 = \pm 1/2$ and $Y_2 = \pm 1$. Considering the large VLL mass limit, the photophobic scenario can be achieved by choosing

$$\frac{y_2}{m_2} = -\frac{y_1}{m_1} , \quad (11)$$

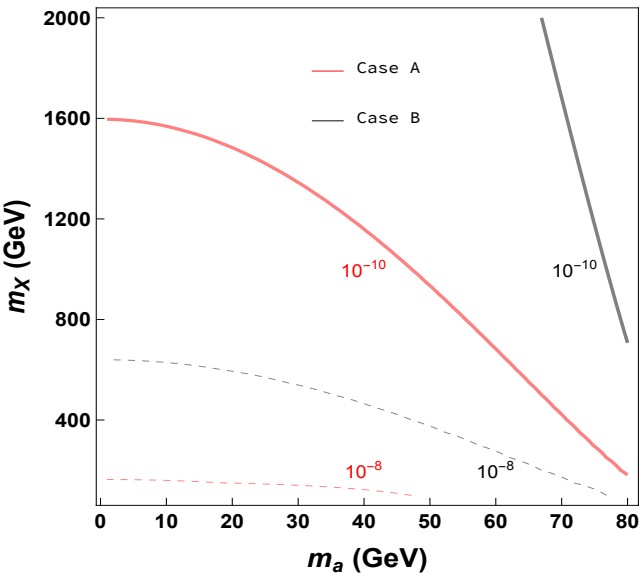

FIG. 1. Contours for two values of BR($Z \to a\gamma$): $10^{-8}$(dotted) and $10^{-10}$(solid) for elementary models. It is shown as a function of the VLL mass and the light pseudoscalar mass. The red (grey) lines correspond to Case A (Case B) respectively. The values are beyond the reach of LEP.

which leads to

$$\frac{C_W}{\Lambda} = -\frac{C_B}{\Lambda} = \frac{1}{32\pi^2} \frac{y_1}{\sqrt{2}m_1} . \quad (12)$$

Note, however, that a coupling of the VLLs to the Higgs can be added in this case, hence mixing the two multiplets and modifying the contribution to the above couplings.

**Case B.** We introduce a triplet $\Psi_1$ and a singlet $\Psi_2$. A photophobic scenario can be facilitated by two choices of hypercharges and the corresponding relation between the Yukawa couplings:
1) If $Y_1 = \pm 1$ and $Y_2 = \pm 1$, we have:

$$\frac{y_2}{m_2} = -5\frac{y_1}{m_1} ;$$

2) If $Y_1 = 0$ and $Y_2 = \pm 1$, we have,

$$\frac{y_2}{m_2} = -2\frac{y_1}{m_1} .$$

Both scenarios lead to the following coefficients

$$\frac{C_W}{\Lambda} = -\frac{C_B}{\Lambda} = \frac{1}{8\pi^2} \frac{y_1}{\sqrt{2}m_1} . \quad (13)$$

In this case, however, a coupling of the VLLs to the Higgs doublet is not possible.

In both cases, the coefficients depend on the ratio of two parameters: the coupling $y_1$ and the VLL mass $m_1$ of one of the two multiplets. These parameters can be tuned to obtained the desired value of the BR($Z \to a\gamma$). Figure

1 illustrates the the contours for two different values of the branching fraction, $10^{-8}$, $10^{-10}$, for $y_1 = 1$ and as a function of the pseudoscalar and VLL masses. The red (black) lines correspond to Case A (Case B), thus ensuring that the pseudoscalar in elementary models will have similar distribution of lifetimes as was noted for the composite analogues [22].

Thus a signature with a monochromatic photon with displaced vertices at the Tera-Z is possible for both composite and elementary models. This leads us towards the consideration of EW precision observables as a possible discriminator.

## IV. PRECISION ELECTROWEAK SIEVE

One of the hallmarks of an electron-positron collider is its capability to achieve high levels of precision. This makes it sensitive to high scales while offering avenues to favour one BSM scenario over the other. In particular, low energy runs at the $Z$ mass and at the $W^+W^-$ threshold, will allow to massively improve the precision measurements in the EW sector of the SM. The possible new physics contributions to various observables can be conveniently parameterised in terms of the oblique parameters $S$ and $T$ [34]. The two ellipses in Fig. 2 show the current 95% CL contours with (green) and without (red) the inclusion of the recent CDF-II measurement of the $W$ mass [35]. The red ellipse corresponds to the fit on the most recent PDG [36], while the new best fit values including CDF-II, $(S, T) = (0.05, 0.15)$, follow from the analyses in [37]. The size of the ellipses reflect the existing levels of precision: the green ellipse is narrower in comparison, as expected, due to the improved precision by the CDF-II measurement. The programme at future $e + e-$ colliders offers to greatly improve the precision on the determination of $S$ and $T$. Due to statistics alone compared to LEP, the precision may improve by a factor of 100, so that the current limiting factor stems from theoretical uncertainties on the SM predictions. A realistic projected precision at FCCee indicates an improvement of $\mathcal{O}(10)$ [38], and we will use this as a benchmark in our analysis.

The oblique parameters have been studied in great detail for both the elementary and the composite models. We present the main features of the impact on these models on oblique parameters before using it for the purpose of discrimination between the elementary and composite light pseudoscalar.

In the elementary mockup model, the simplified cxSM, there are two contributions to the oblique parameters: A) the mixing between the scalar singlet component to the SM Higgs boson, and B) the contribution to the vacuum polarisation due to the VLLs. In particular, the pseudoscalar $a$ does not directly contribute. The contribution

of the VLLs to the S parameter is given as

$$\Delta S_{\text{VLL}} = -\frac{4}{3\pi} \sum_j \text{Tr} \left[ T^3 Y_j \log \left( \frac{m_{\Psi_j}^2}{\Lambda^2} \right) \right] , \qquad (14)$$

where the trace acts on the components of the multiplets $\Psi_j$. This contribution vanishes unless mass differences among multiplet components are induced, which can only come from Yukawa couplings to the Higgs. The same applies to $\Delta T_{\text{VLL}}$, as violation of the custodial symmetry can only come from Yukawa couplings of the Higgs doublet. Hence, in the most minimal scenarios, the VLLs do not contribute. The only non-vanishing effect comes from the mixing of the singlet scalar $S$ with the Higgs from the doublet, and it is sensitive to the mixing angle $\phi$, proportional to $\delta_2$ in the scalar potential. The contribution to the oblique parameters due to this mixing is given as

$$\Delta T_\phi = -\frac{3}{8\pi c_W^2} \ \sin^2\phi \ \log\frac{m_{h_2}}{m_{h_1}},$$
$$\Delta S_\phi = \frac{1}{6\pi} \ \sin^2\phi \ \log\frac{m_{h_2}}{m_{h_1}} , \qquad (15)$$

where $m_{h_1}$ is identifies as the 125 GeV Higgs candidate while $m_{h_2}$ is the mass of the (heavier) CP-even eigenstate. Hence, the oblique observables and the production dynamics of the pseudoscalar depend on two non-overlapping and independent sets of parameters: $\{\phi, m_{h_2}\}$ and $\{m_a, m_{\text{VLL}}\}$, respectively. As a result, the observation of the $Z \to a\gamma$ process with a given BR does not lead to a precise prediction for the $(S, T)$ values. Nevertheless, as the contributions are proportional to each other, the cxSM model can only lie on a line, shown as the black solid line in Fig. 2. As already mentioned, additional contributions may come from the VLLs, if a coupling to the Higgs doublet $H$ is present and sizeable.

This is in stark contrast to the scenario in composite models. The oblique parameters, in this latter case, can be expressed as [39]

$$\Delta T_{\text{FC}} = -\frac{3}{8\pi c_W^2} \ \frac{v^2}{f^2} \ \log\frac{\Lambda_{FC}}{m_h} ,$$
$$\Delta S_{\text{FC}} = \frac{1}{6\pi} \ \frac{v^2}{f^2} \ \left( \log\frac{\Lambda_{FC}}{m_h} + N_D \right) , \qquad (16)$$

where $\Lambda_{FC} \approx 4\pi f$ is the condensation scale of the underlying fundamental gauge (FC) theory and $N_D$ counts the number of chiral $SU(2)_L$ doublets in the confining theory. For minimal cosets, $N_D = d_\psi$, c.f. Eq. (2). Note that the contributions proportional to the logarithm come from modifications of the Higgs couplings and are, therefore similar to the cxSM terms in Eq. (15). The term proportional to $N_D$, instead, estimates the non-perturbative contribution of the confining sector, which is assumed to be custodial invariant and hence not contributing to $T$. Contrary to the elementary case, $(S, T)$ in Eq. (16) can be related to the $Z \to a\gamma$ process via $f$, which can be

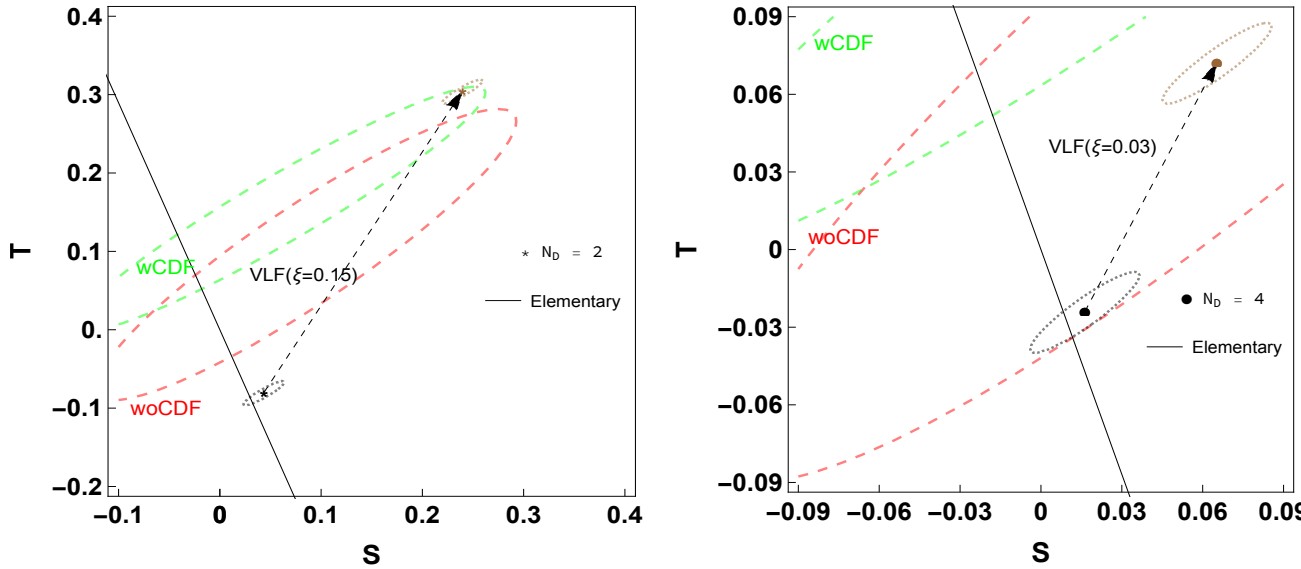

FIG. 2. A comparison of the contributions to the oblique parameters between the composite and the elementary models. The red and green curves represent the current 95% regions with and without the inclusion of CDF-II $W$ mass result, respectively. The solid black line is the elementary contribution. The dots show the contribution of the composite models without (black) and with (brown) the top partners. We fixed $\mathrm{BR}(Z \to a\gamma) = 10^{-8}$ and small $m_a$, showing results for $N_D = 2$ (left) and $N_D = 4$ (right). The small grey ellipses estimate the errors after future $e + e-$ colliders as $1/10$ of the ones of the green ellipse.

obtained as a function of the BR by inverting Eq. (4). Hence, once $m_a$ and BR are fixed by the discovery at the Tera-Z, $(S, T)$ are fully fixed in terms of $N_D = d_\psi$. In Fig. 2 we show two examples for BR= $10^{-8}$ and small $m_a$, with $N_D = 2$ (left plot) and $N_D = 4$ (right plot). We can already see that the prediction for the composite models are very close to the mockup model like, hence it will be hard to distinguish them unless the precision on the EW observables is greatly enhanced, via more accurate theoretical calculations. As a reference, in the plot we draw an ellipse centred on the composite prediction, based on the green contour but with errors reduced by a factor of 10. Note that smaller BR would require larger $f$, hence proportionally smaller contributions to $S$ and $T$.

The central value predicted by composite models also lies well outside of the current ellipses, hence it is strongly disfavoured, especially after the inclusion of the CDF-II measurement of the $W$ mass (green ellipse). However, complete composite Higgs models feature top partial compositeness [40], characterised by the presence of potentially light fermionic top partners [41, 42]. As a result, additional contributions to the oblique parameters are generated, which could push the black points in Fig. 2 closer to the experimental ellipse and away from the mockup model values. A simple estimate of the contribution of top partners is given as [41]

$$\Delta T_{TP} \simeq \frac{3}{16\pi^2} \, y_t^2 \, \xi \,,$$
$$\Delta S_{TP} \simeq \frac{g^2}{8\pi^2} \left(1 - 2c_\theta^2\right) \, \xi \, \log\left(\frac{m_*^2}{m_4^2}\right) \,, \quad (17)$$

where $y_t \sim 1$ is the SM top Yukawa, $\xi = \frac{v^2}{f^2}$, $c_\theta$ is the co-

sine of a mixing angle between the top and the composite top partners, and $m_\star/m_4 \sim 1$ is the ratio of masses in the top partner sector, generated by the strong dynamics. The key message is that a largish and positive contribution is expected for $T$, while the effect on $S$ is somewhat smaller. Also, $\Delta T_{TP}$ is completely fixed by $f$, which is fixed as before as a function of the BR and $N_D$. The additional contribution to $(S, T)$ from top partial compositeness is indicated schematically by the arrows in Fig. 2, where the length of the shift is fully determined in the $T$ direction. These contributions tend to push the composite models towards the experimental ellipse while also disentangling them from the mockup elementary model.

## V. RESULTS AND CONCLUSIONS

The presence of light composite pseudoscalars offers the possibility of using future low energy $e + e-$ colliders at the $Z$ resonance as discovery machines for compositeness. The new state can be discovered in the channel $Z \to a\gamma$ and prove high compositeness scales, as demonstrated in Ref. [22]. In this letter we have demonstrated that electroweak precision measurements, which will also be performed at the same machine, can potentially distinguish the composite pseudoscalar from elementary mockups.

As shown in Fig. 2, the projected precision, which will improve by a factor of $1/10$ with respect to the current fits, is not enough *per se* to disentangle the two models. However, contribution from top partial compositeness play a crucial role in pushing the composite predic-

tion in the $(S, T)$ plane away from the mockup and also closer to the currently allowed ellipse. It should be remarked that we did not include the effect of vector-like leptons in the mockup model, as effects on $S$ and $T$ can only be generated if couplings of the Higgs are included.

In the composite model, there is a remarkable correlation between the value of the BR($Z \to a\gamma$) and $(S, T)$, which can be deployed to uncover some details of the underlying confining dynamics. Instead, in the mockup elementary model, this correlation is not present. Nevertheless, achieving the explored signals required the presence of relatively light states: vector-like leptons below a few TeV and a second Higgs boson. The former can be searched for at a high energy hadron collider, while the presence of the latter will be tested via precision measurement of the Higgs couplings at the High-Luminosity

LHC and future $e + e-$ colliders in their Higgs-factory run. Hence, this simple model offers a rich physics case for a variety of future collider projects.

## ACKNOWLEDGEMENTS

We acknowledge support from CEFIPRA under the project "Composite Models at the Interface of Theory and Phenomenology" (Project No. 5904-C). AI would like to thank IP2I Lyon for the hospitality where the project was initiated. We also acknowledge the support of the FCC-France Initiative for financing the internship of AP at IP2I.

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
