# Peer review of "Sifting composite from elementary models at FCCee and CePC"

_SciPost Physics_

## Round 1 · Referee Report · Anonymous (Referee 1) · 2022-12-20

Report
The paper discusses the WZW interaction and induced Z boson decays as a probe of compositeness. Unfortunately, I have to say that I find the paper somewhat confusing. The BR in Eq.(4) appears to be a dimensionful quantity, the potential in Eq. (5) is not real (probably lacking "h. c."s somewhere). I can believe that these are typos (or perhaps I'm missing something), however, at this level it is not clear to me if the additional parts are impacted.
The physics that is discussed in the paper is no new and relatively straightforward. Technical complexity is not requirement for good ideas, but I don't see how this work critically adds to a plethora of articles in the literature that already discuss the combination of electroweak precision and exotic decays as a source of electroweak insights, irrespective of whether they are of composite or elementary nature.
The physics that is discussed in the paper is no new and relatively straightforward. Technical complexity is not requirement for good ideas, but I don't see how this work critically adds to a plethora of articles in the literature that already discuss the combination of electroweak precision and exotic decays as a source of electroweak insights, irrespective of whether they are of composite or elementary nature.

Author: Abhishek Iyer on 2023-02-07 [id 3329]
(in reply to Report 1 on 2022-12-20)We present a point wise reply to the different aspects flagged by the referee:
A) Referee: The BR in Eq.(4) appears to be a dimensionful quantity,
Our Reply: The expression in Eq. 4 is indeed a typographical error. The correct expression is:
\begin{equation}
\mbox{BR} = \frac{8 \pi \alpha\ m_Z^3}{3 s_W^2 c_W^2 \Gamma_Z} \left( 1-\frac{m_a^2}{m_Z^2} \right)^3 \ \left( \frac{C_W}{\Lambda} \right)^2\,.
\end{equation}
This will be corrected in the next version.
B) Referee: the potential in Eq. (5) is not real.
Our Reply: This is also a typographical error and an "h.c" will be included in the next version.
These two typos do not affect the results presented in the paper.
C) Referee: The physics that is discussed in the paper is no new and relatively straightforward. Technical complexity is not requirement for good ideas, but I don't see how this work critically adds to a plethora of articles in the literature that already discuss the combination of electroweak precision and exotic decays as a source of electroweak insights, irrespective of whether they are of composite or elementary nature.
Our Reply: We wish to point out that the purpose of the paper is to develop the connection between electroweak precision and exotic decays in a specific composite Higgs set-up with light resonances. Such connection was never considered before. In fact it builds on previously established ideas to demonstrate the discriminating power of electroweak precision variables to distinguish between elementary and composite models. This is again a new analysis compared to what already present in the literature. It must be noted that the underlying mechanisms which lead to such decays leave different imprints on electroweak precision observables. This is the main thrust of the paper. The results are shown in Fig. 2 of the manuscript and serve as a useful benchmark for the development of FCCee strategy in terms of desired precision goals and hence it is an important contribution in that direction.

---

## Editorial Decision

resubmitted